# Examination of Microcystin Adsorption by the Type of Plastic Materials Used during the Procedure of Microcystin Analysis

**DOI:** 10.3390/toxins14090625

**Published:** 2022-09-07

**Authors:** Chan Seo, Joo Won Lee, Won-Kyo Jung, Yoon-Mi Lee, Seungjun Lee, Sang Gil Lee

**Affiliations:** 1Department of Food Science and Nutrition, Pukyong National University, Nam-Gu, Busan 48513, Korea; 2Department of Smart Green Technology Engineering, Pukyong National University, Busan 48513, Korea; 3Research Center for Marine Integrated Bionics Technology, Pukyong National University, Busan 48513, Korea; 4Department of Biomedical Engineering, Pukyong National University, Busan 48513, Korea; 5Food Safety and Processing Research Division, National Institute Fisheries Science, Busan 46083, Korea

**Keywords:** microcystin, adsorption, plastic, polyethylene terephthalate, regenerated cellulose

## Abstract

The incidence of eutrophication is increasing due to fertilizer abuse and global warming. Eutrophication can induce the proliferation of cyanobacteria such as *Microcystis*, which produces microcystins. Microcystins are toxic to specific organs such as the liver and the heart. Thus, monitoring of microcystins is strongly required to control drinking water and agricultural product qualities. However, microcystins could be adsorbed by plastic materials during sample storage and preparation, hindering accurate analysis. Therefore, the current study examined the recovery rate of microcystins from six plastics used for containers and eight plastics used for membrane filters. Among the six plastics used for containers, polyethylene terephthalate showed the best recovery rate (≥81.3%) for 48 h. However, polypropylene, polystyrene, and high- and low-density polyethylenes showed significant adsorption after exposure for 1 hr. For membrane materials, regenerated cellulose (≥99.3%) showed the highest recovery rate of microcystins, followed by polyvinylidene fluoride (≥94.1%) and polytetrafluoroethylene (≥95.7%). The adsorption of microcystins appeared to be strongly influenced by various molecular interactions, including hydrophobic interaction, hydrogen bonding, and electrostatic interaction. In addition, microcystins’ functional residues seemed to be critical factors affecting their adsorption by plastic materials. The present study demonstrates that polyethylene terephthalate and regenerated cellulose membrane are suitable plastic materials for the analysis of microcystins.

## 1. Introduction

The incidence of eutrophication due to environmental pollution and fertilizer abuse is increasing [1,2,3,4,5]. Eutrophication causes a massive proliferation of toxin-producing algae such as cyanobacteria, which harms animals and plants [6,7,8,9]. Among cyanobacteria, *Microcystis* produces microcystins (MCs), which have hepatotoxicity [10,11]. In general, MCs consist of seven amino acids, including the Adda structure separated by two variable residues. About 279 different MC congeners have been reported [12,13]. MCs are very harmful when ingested or inhaled. Humans are exposed to MCs by consuming foods grown with water contaminated by MCs and inhalation in underwater recreation [14,15]. For example, cases of MCs exposure in Brazil have been reported [16,17]. Massive acute exposure and chronic exposure to low concentrations of MCs can cause toxicity by accumulating in the liver, kidney, and heart [18,19,20,21,22]. Among various MC congeners, MC-LR is known to be the most toxic compound. The structure of MC-LR with modified amino acids of leucine and arginine shows high resistance to heat, hydrolysis, and oxidation [23]. Due to such toxic effects of MCs, monitoring concentrations of MCs in food ingredients and drinking or agricultural water is very critical [21,24,25,26]. The WHO recommended a concentration of MCs in drinking water that is less than 1.0 μg/L for chronic exposure and 12 μg/L for short-term exposure [27].

In general, the analysis of MCs is performed using liquid chromatography (LC) combined with UV/visible detector (UVD) or mass spectrometry (MS) [28]. To have accurate analytical results using HPLC with detectors, it is critical to remove various impurities through filtering, extracting (solid-phase or liquid-phase), or concentrating [29]. However, it has been reported that MC-LR can be adsorbed by various types of plastics such as polystyrene (PS), polypropylene (PP), and polyvinyl chloride (PVC) [30,31,32,33]. Since many laboratory tools are made of plastics, MCs will be exposed to plastics inevitably during the sample preparation for the analysis of MCs. The loss of MCs during analysis sample preparation can cause significant problems in accurate quantitation of MCs [31,32]. It has been reported that methanol could improve the recovery rate of MC-LR by reducing MC-LR adsorption into plastic materials [31]. However, for MC recovery, adding an organic solvent such as methanol can dilute the MC in water. In addition, it requires additional time and preparation steps such as a concentrating step. Therefore, selecting the appropriate plastic material for sample preparation is very important. However, little research has been conducted on the impact of adsorption loss caused by various plastic materials. In particular, information on adsorption of MCs on the syringe filter membrane, which is essential for analysis of MCs using LC, is insufficient. Therefore, the objective of this study was to thoroughly examine adsorption rates of MC-LR, -RR, and -YR by different plastic materials used for plastic containers and syringe filters. 

## 2. Results

### 2.1. Method Validation for Microcystins

Method validation for analysis of three MCs in distilled water was performed using a published method [34] with slight modifications. Under optimal conditions, the retention time of MC-LR, -RR, and -YR ranged from 3.79 to 4.11 min (Appendix A). In the concentration range from 1 µg/L to 20 µg/L, MC-LR, -RR, and -YR showed good linearity (R^2^ ≥ 0.999), with limits of detection ranging from 0.001 µg/L to 0.002 µg/L, and quantification from 0.002 µg/L to 0.005 µg/L. Accuracy (percent relative error) and repeatability (relative standard error) were −3.4 to 0.7% and 3.3 to 6.2%, respectively (Table 1).

### 2.2. Adsorption Effects of Six Plastic Containers on Analysis of Microcystins

Recovery rates of MC-LR, -RR, and -YR after exposure to plastic particles for 0, 1, 6, 12, 24, and 48 h were investigated using six different types of plastic particles (PET, PP, PFA, PS, HDPE, and LDPE) (Figure 1A for -LR, Figure 1B for -RR, and Figure 1C for -YR). After exposure for 48 h, PET (Figure 1A, Black Circle) showed the highest MC-LR recovery rate of 91.5%. PFA (Figure 1A, Black Triangle) showed a considerable recovery rate of 75.5% at 6 h. However, its recovery rate dramatically dropped to less than 21.8 at 12 h. In the case of PS (Figure 1A, White Circle), the recovery rate of MC-LR was about 60% at 6 h and less than 18.5% after 12 h. Among the six plastic particles, LDPE (Figure 1A, White Triangle) and HDPE (Figure 1A, Black Diamond) showed the most significant reduction in the recovery for the three MCs. LDPE and HDPE particles resulted in less than 45.4% of recovery rate at 1 h and a recovery rate of 18.5% after 12 hr. In the case of PP (Figure 1A, Black Square), the recovery rate of MC-LR dropped significantly (39.5%) by 1 h. This recovery rate lasted until 24 h exposure. It decreased to less than 20% after 48 h exposure. 

Concerning MC-RR and -YR, their recovery rates showed a similar trend to recovery rates of MC-LR after exposure to the six plastic particles. PET (Figure 1B) was still a desirable material, resulting in a recovery rate of 81.6% after 48 h exposure. PFA (Figure 1B) resulted in a recovery rate of 67.7% after exposure for 6 h. The recovery rate was sharply decreased after 12 h of exposure. It was only 6.9% after 48 h of exposure. MC-RR exhibited similar recovery rates after exposure to PP, PS, HDPE, and LDPE (Figure 1B). After exposure to these four plastic particles (PP, PS, HDPE, and LDPE), over 59.8% of MC-RR were adsorbed after 1 h of exposure. The adsorption was then gradually increased after exposure for 48 h. The best MC-YR recovery rate was found when it was exposed to PET (Figure 1C). Its recovery rate remained at 81.3% after 48 h exposure. The MC-YR recovery rate after exposure to PFA (Figure 1C) showed a similar trend to the MC-LR and MC-RR recovery rates. After 6 h exposure, the recovery rate of MC-YR was 75.3%. After 12 h exposure, its recovery rate rapidly decreased from 31.1% to 8.3%. After exposure to PS for 1 h (Figure 1C), the recovery rate dropped to 53.8%. The recovery rate was maintained at a similar level (51.8%) for 6 h. After 12 h, the recovery rate of MC-YR dropped from 13.5% to undetectable levels. After exposure to PP, the recovery rate of MC-YR was decreased to 35.7%. This rate was maintained until 24 h. However, about 90% of MC-YR was lost after 48 h of exposure. The recovery rate of MC-YR exposed to LDPE (Figure 1C) and HDPE (Figure 1C, filled diamond) was decreased significantly to 38.8% after 1 h exposure. It was gradually decreased to 11.0% after 48 h exposure. A similar trend in adsorption of the three MCs on the six plastic materials was observed (Appendix A).

### 2.3. Effects of Plastic Materials Used for Eight Membrane Filters on Adsorption of Microcystins

Effects of different membranes on the recovery rates of MC-LR, -RR, and -YR were determined after filtering 500 μL of standard solution (10 ng/mL in final) using eight different types of membrane syringe filters (RC, PVDF, PTFE, CA, MCE, PES, NY, and PP) (Figure 2). Among these eight membrane types, RC resulted in the highest recovery rate (≥99.3%) of MCs, followed by PTFE (≥95.7%) and PVDF (≥94.1%) (Figure 2A). In addition, RC, PTFE, and PVDF resulted in a similar trend of recovery for the three MCs. On the other hand, filtration using CA, MCE, NY, PES, and PP significantly decreased the recovery rates of the three MCs (Figure 2A). The MC-LR and -YR recovery rates using CA membrane filters were 92.9% and 95.0%, respectively. The recovery rate for MR-RR using the same CA filter was slightly reduced to 88.1%. With MCE membrane filters, MC-LR, -RR, and -YR showed relatively lower recovery rates of 91.4%, 79.9%, and 88.3%, respectively. NY, PES, and PP filters were found to be unsuitable for accurate measurements. When the NY filter was used, MC-RR showed a reasonable recovery rate of 99.1%. However, MC-YR and -LR showed the worst recovery rates of 37.7% and 67.7%, respectively (Figure 2B). In the case of the PES filter, MC-LR, -RR, and -YR showed recovery rates of 82.2%, 80.1%, and 61.4%, respectively (Figure 2B). Among the eight syringe filters, PP resulted in the lowest recovery rate and the most significant loss of MCs, leading to a recovery rate of 51.1% for -LR, 63.6% for -RR, and 51.8% for -YR (Figure 2A).

## 3. Discussion

As MC-LR in food and drinking water causes hepatotoxicity such as intrahepatic bleeding when it is administered orally, monitoring of MCs is essential for public health [35]. However, MC-LR is adsorbed by PP or PVC used for sample pretreatment, which may cause problems in accurate monitoring [32,36]. Nevertheless, few research studies have been conducted on the impact of adsorption loss by different plastic materials. In particular, studies about the adsorption effects of membrane materials are scarce. Adsorption of an organic compound to plastics can be affected by the plastics’ glassiness, crystallinity, and polarity [37,38]. Glass transition temperature (*T*_g_) is a critical factor affecting glassy or rubbery characteristics of plastics. When plastics are under higher temperatures than their glass transition temperatures, they can keep an elastic state called the rubber state [39]. The rubber state contains more amorphous properties than the glassy state by which the plastic can adsorb more organic compounds, as intermolecular bonds are more flexible in the rubber state than in the glass state [40,41]. PE is generally considered a rubbery material, while PET is considered a glassy plastic [42]. In a previous study, PE showed higher absorption rates of organic compounds than PET [32,33,39]. Another study has reported that organic compounds have low permeability and adsorption rates by structurally aligned plastics (high crystallinity) [41]. 

The adsorption of organic compounds can be explained by chemical features such as hydrophobicity or hydrophilicity of functional groups [39,42]. Li et al. [43] have reported that hydrophobic antibiotics have higher affinities for PP, PS, and PE. Therefore, interactions between the plastic and two variable amino acids of MCs might affect the absorption capacity. Plastic adsorption rates for MC-LR, -RR, and -YR were compared using six containers (Figure 3A). Recovery rates of MC-LR, -RR, and -YR after exposure to plastic particles (400–800 μm) for 48 h are as follows: PET > PFA > PS > PP > LDPE > HDPE > HDPE. PET contains an ester group with a strong electronegativity. Ester groups can form strong bonds through electrical interactions. Such chemical characteristics of PET could have a strong bonding capacity within polymers of PET main chains [33,44,45]. Thus, strong bonding between polymers could minimize plastics’ bonding capacity for MC-LR, -RR, and -YR, providing excellent recovery rates (over 81% after 48 h). PFA is a plastic made of carbon-fluorine bonds with high thermal and chemical stability. Thus, it is useful for storing various chemicals. However, when MC-LR, -RR, and -YR were exposed to PFA (Figure 1) for 1 h, the recovery rates for MC-LR, -RR, and -YR were confirmed to be ≥ 75.3%, which were higher than those when they were exposed to four other plastic particles, except for PET. However, the recovery rates were dramatically reduced to ≤31.1% after 12 h. Although the C–F dipole of PFA is capable of electrostatic interactions with other dipoles or charges, the PFA is very stable with a low reactivity [46]. PP (Figure 1) was one of the plastics that resulted in poor recovery of MC-LR, -RR, and -YR. The three MCs used in the current study were adsorbed by PP to about 87% during 48 h of exposure. In previous studies on organic compounds such as antibiotics, PP has a higher adsorption capacity than PET, which is explained by the rubbery state, crystallinity, and van der Waals interactions [47,48,49,50]. Plastic with high crystallinity or that is glassy is expected to show low adsorption of MCs. PP is generally known as a rubbery material with a low crystallinity. However, PP showed higher MC recovery rates than PS and PE with high crystallinity [43]. This result indicates that various factors as well as the physicochemical properties of plastics could affect the adsorption rates of MCs. PS (Figure 1) showed a similar recovery rate to PE for 12 h, although the recovery rate was decreased to undetectable levels. The large adsorption capacity of PS might be due to π (Pi) interaction by aromatic rings [39,43,51]. Therefore, it is assumed that the adsorption of plastic is mainly due to van der Waals interactions and hydrophobic or π interactions of hydrocarbon chains as well as phenyl groups, rather than properties such as rubber state or crystallinity.

Syringe filtration is an essential process for analysis using LC. The structure of the housing system and the membrane of the syringe filter are similar to plastics. Thus, loss of MCs due to adsorption should be considered. The recovery rates of MC-LR, -RR, and -YR after exposure to syringe filters were investigated using seven filters, including RC membranes (Figure 3B). RC (≥99.3%), PVDF (≥94.1%), and PTFE (≥95.7%) were confirmed to be the best syringe filters for the recovery of the three MCs (Figure 2A). PVDF and PTFE are fluoropolymers like PFA. Membrane exposure after filtration was relatively short. PVDF and PTFE showed reasonable recovery rates at ≥94.1%. PVDF and PTFE contain solid carbon-fluorine bonds, which have a low space for chemically interacting with other compounds. CA is a membrane made by acetylation of cellulose. A loss of about 12% was observed for MC-RR due to adsorption by CA (Figure 2A). The loss of three MCs in filtration using CA is thought to be due to the two amino acids in variable residues of MC [33]. MC-LR, -RR, and -YR differ in two variable residues consisting of leucine (Leu, L), arginine (Arg, R), and tyrosine (Tyr, Y), which include an amine group or hydroxyl group with high hydrophilicity [52,53]. These characteristics of variable residues (Arg and Tyr) are more adsorbed due to hydrogen bonding and electrostatic interaction in the storage container [54,55,56]. The surface of the hydrophilic CA membrane contains an amine group and an acetyl group capable of hydrogen bonding or electrostatic interaction, which can account for the loss of the three MCs. Based on this mechanism, it is possible to explain the low recovery rate of MC-RR composed of two Arg. This adsorption mechanism by functional groups of variable residues can also be applied to MCE and PES. MCE is composed of cellulose nitrate and cellulose acetate. PES is an aromatic polymer consisting of two aromatic rings and a sulfone group (Figure 3B). MCE has little effect on hydrogen bonding, because the hydroxyl group is substituted with nitrate and acetate. Adsorption due to electrostatic interaction might be proposed. PES’s sulfonic groups and aromatic rings can exhibit electrostatic interactions, n-π interactions, and π-π interactions [57]. It suggests a high adsorption potential of hydrophilic compounds. In particular, the (all-S, all-E)-3-amino-9-methoxy-2,6,8-trimethyl-10-phenyldeca-4,6-dienoic acid) structure and Tyr of MC-YR, which are related to π interactions, are expected to be involved in the adsorption. It is consistent with MC-YR showing the lowest recovery (61.4%) after exposure to PES (Figure 2B) [43]. On the other hand, although RC has a similar structure to MCE and CA, RC resulted in a much better recovery (≥99.3%) than the other two. The cellulose of RC is crystalline with a high structure order. It has strong intermolecular interactions such as hydrogen bonds [58,59]. Thus, it is considered that RC has a low adsorption rate due to hydrophilicity and high chemical stability. NY membrane has a high adsorption capacity for proteins. It is known that acidic protein is adsorbed more than basic protein. This NY characteristic can be explained that MC-RR is composed of two arginines. Thus, MC-RR showed a lower adsorption rate than MC-LR and -YR (Figure 2B). This mechanism can explain the low recovery rate of MC-YR containing relatively acidic Tyr [60]. Hyenstrand et al. [32] have reported that MC-LR is quickly adsorbed by the PP disposable pipette tip in an aqueous solution. Among membranes used in the present study, PP resulted in a low recovery rate (≤ 63.6%) in a short time. PP also resulted in rapid and significant losses (≤ 39.5%) in comparison with other plastic particles of storage containers (Figure 1). Thus, it was considered that the absorption of the three MCs by PP was due to the hydrophobic interaction between the highly hydrophobic hydrocarbon polymer and MCs [61]. This is consistent with results showing that the MC-RR consisting of two Arg with high hydrophilicity has a higher recovery rate than MC-LR and -YR. Therefore, the adsorption of MC-LR, MC-RR, and -YR by membranes is considered to be mainly due to hydrophobic interactions and chemical properties of amino acids in variable residues. The adsorption of MC was investigated using particles of six plastic containers. This means that MC adsorption might be different in normal plastic containers. In addition, he confirmed results of interactions between the three MCs and plastic particles provide useful information to select suitable materials for plastic containers and filters.

## 4. Conclusions

The effects of plastic materials of six storage containers and eight membrane syringe filters on the recovery rates of MC-LR, -RR, and -YR were investigated. PET and RC membranes were the most desirable filter membranes for storing and analyzing MC-LR, -RR, and -YR. The adsorption rate of MCs tends to be influenced by hydrophobic interaction, hydrogen bonding, and electrostatic interaction. Different residues of MC congeners resulted in various chemical properties. Therefore, the material type of the storage container and membrane filter should be considered when performing simultaneous analysis of MC congeners. 

## 5. Materials and Methods

### 5.1. Chemicals and Materials

Microcystins (RR, LR, and YR, ≥95.0%, 10 µg/mL in methanol, analytical standard) and formic acid were purchased from Sigma-Aldrich (St. Louis, MO, USA). HPLC-grade water and acetonitrile were purchased from Fisher Scientific (Ottawa, Canada). Regenerated cellulose (RC, 0.2 μm), polyether sulfone (PES, 0.2 μm), and nylon (NY, 0.2 μm) membrane syringe filters were purchased from Sartorius (Darmstadt, Germany). Polyvinylidene fluoride (PVDF, 0.2 μm), polytetrafluoroethylene (PTFE, 0.2 μm), mixed cellulose esters (MCE, 0.2 μm), and polypropylene (PP, 0.2 μm) membrane filters were purchased from JET BIOFIL (Guangzhou, China). Cellulose acetate (CA, 0.2 μm) membrane syringe filters were purchased from GVS (Sanford, ME, USA). Polystyrene (PS) and polypropylene (PP) tubes were purchased from SPL Life Science (Gyunggi-do, Korea). Low-density polyethylene (LDPE) and high-density polyethylene (HDPE) bottles were purchased from Thermo Fisher Scientific (Lexington, KY, USA). The sample bottle of polyethylene terephthalate (PET) was purchased from YD platech (Gyunggi-do, Korea). Polyfluoro alkoxy (PFA) was purchased from Lklab (Seoul, Korea).

### 5.2. Preparation of Standard Solution

Standard stock solutions for MC-LR, -RR, and -YR were prepared at 10 µg/mL in methanol. Working solutions for mixed MC standards were made by mixing and diluting with a calibrated micropipette at 1 µg/mL in distilled water. All standard solutions were stored at −20 °C.

### 5.3. Adsorption of Microcystins on Different Plastic Materials of Storage Containers

Six different types of plastic particles, including PS, PP, LDPE, HDPE, PET, and PFA from plastic bottles and tubes, were prepared by grinding these bottles and tubes (Figure 3A) using a steel grinder. Ground plastic particles were then filtered through a 400 μm sieve to remove small particles. Subsequently, the residue was filtered using 800 μm sieves to prepare plastic particles with sizes ranging from 400 μm to 800 μm. Briefly, 30 mg of the plastic particle were added to a 1.5 mL glass vial. Then, 500 μL (5 μL of working solution in methanol and 495 μL of distilled water) of mixed MC standard solution were added (final concentration at 10 ng/mL) to induce adsorption of MC-LR, -RR and -YR by plastic particles using a rotator shaker (SeouLin Bioscience, Seongnam, Gyeonggido, Korea) at 80 rpm for 1, 6, 12, 24, and 48 h at room temperature, respectively. After shaking, mixed samples were centrifuged at 13,500 rpm for 1 min. Subsequently, 50 μL of supernatant were used for LC-MS/MS analysis of adsorption rates for MCs. The experiment was conducted in triplicate. The results are presented as mean ± standard deviation (SD).

### 5.4. Adsorption of Microcystins by Different Plastic Materials of Membrane Syringe Filters

The effects of membrane syringe filters on the adsorption rate of MCs were investigated using eight different types of membranes, including RC, NY, PVDF, PTFE, MCE, PP, and CA (Figure 3B). 500 μL of a mixture of MCs (final concentration at 10 ng/mL) were filtered with each filter using a glass syringe. Filtrates were used for LC-MS/MS analysis. The experiment was conducted in triplicate. The results are presented as mean ± SD.

### 5.5. Microcystin Analysis Using UPLC-MS/MS

A Xevo TQ-MS triple quadrupole mass spectrometer (Waters, Guyancourt, France) equipped with a Waters Acquity UPLC system (Waters, Guyancourt, France) was used to analyze the MC recovery rate after exposure to different plastic materials. Chromatographic separation was achieved on a Waters BEH C18 column (2.1 mm × 100 mm of inner diameter, 1.7 μm of particle size; Waters, Guyancourt, France). The analytical condition was as follows: electrospray ionization with positive mode, desolvation temperature of 500 °C, desolvation gas flow rate of 700 L/h, and source temperature of 150 °C. Distilled water containing 0.1% formic acid (A) and acetonitrile containing 0.1% formic acid (B) were used for mobile phases. The gradient condition was as follows: 0–1 min, 5% (B); 1–5 min, 100% (B); 5–7.5 min, 100% (B); and 7.5–10 min, 5% (B). The column was maintained for 2.5 min with 95% (A). MC-LR, -RR and -YR were monitored in multiple reaction monitoring mode under optimized conditions as follows: MR-LR: *m/z* 995.4 > 134.9 and 106.95; MR-RR: *m/z* 520.0 > 134.9 and 106.9; and MR-YR: *m/z* 1045.4 >134.9 and 106.9.

### 5.6. Statistical Analysis

The experiment was conducted in triplicate. The results are presented as mean ± SD. One-way analysis of variance (ANOVA) and Tukey’s post hoc test were performed using GraphPad Prism version 9.00 (San Diego, CA, USA). Significant differences between the results were analyzed using a Tukey post hoc test at a significance level of *p* < 0.05.

## Figures and Tables

**Figure 1 toxins-14-00625-f001:**
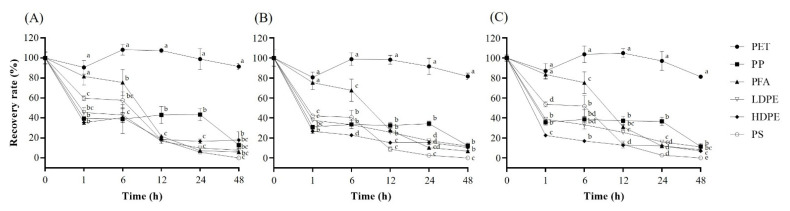
Effects of six plastic materials including polystyrene (PS), polypropylene (PP), low-density polyethylene (LDPE), high-density polyethylene (HDPE), polyethylene terephthalate (PET) and polyfluoro alkoxy (PFA) used for storage containers on recovery rates of (**A**) MC-LR, (**B**) MC-RR, and (**C**) MC-YR. The recovery rate was calculated as the peak area of standard solution exposed to plastic container material or membrane filter/peak area of pure standard solution ×100. Different letters (a–e) indicate significant difference (*p* < 0.05).

**Figure 2 toxins-14-00625-f002:**
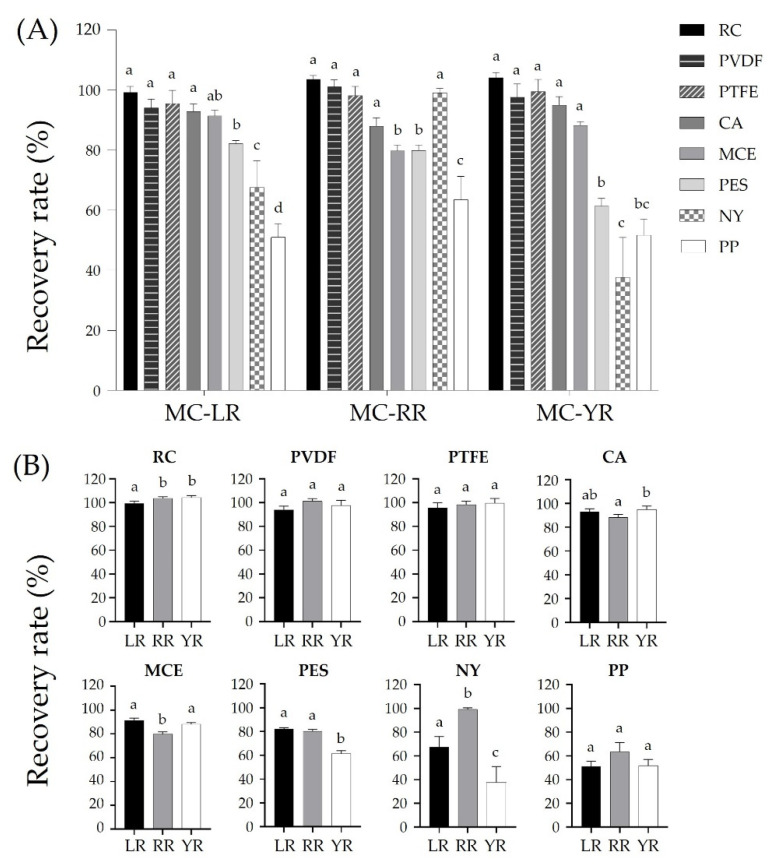
Recovery rates of three MCs. (**A**) Effects of eight membrane filters on recovery rates of the given three MCs; (**B**) Recovery rates of the three MCs with a given membrane filter. RC, regenerated cellulose; PVDF, polyvinylidene fluoride; PTFE, polytetrafluoroethylene; CA, cellulose acetate; MCE, mixed cellulose esters; PES, polyether sulfone; NY, nylon; PP, polypropylene. Recovery was calculated as the area of peaks of standard solutions after filtration divided by the area of peaks of standard solutions before filtration × 100. Different letters (a–d) indicate significant difference (*p* < 0.05).

**Figure 3 toxins-14-00625-f003:**
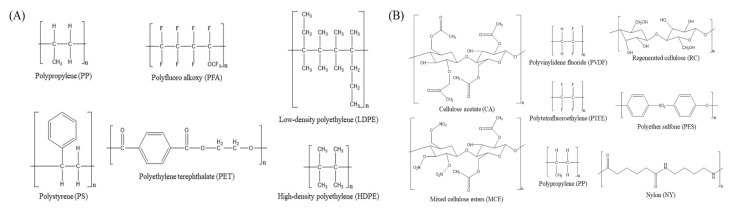
Chemical structures of plastic materials used in the current research. (**A**) Plastic materials for bottles and tubes, (**B**) Plastic materials for membranes of syringe filters.

**Table 1 toxins-14-00625-t001:** Multiple reaction monitoring conditions and validation data set for MC-LR, -RR, and -YR.

Analyte	RT	Ionization Mode	Precursor Ion	Product Ion	Calibration Range (μg/L)	Linearity ^a^ (R^2^)	LOD	LOQ	Accuracy ^b^ (%)	Repeatability ^c^ (%)
			(*m/z*)				μg/L			
MC-RR	3.79	+	520.0	134.9	1–20	0.999	0.001	0.004	0.7	6.2
MC-LR	4.07	+	995.4	134.9	1–20	0.999	0.001	0.002	−3.4	8.5
MC-YR	4.16	+	1045.4	134.9	1–20	0.999	0.002	0.005	−2.9	3.3

RT, Retention time; LOD, Limit of detection; LOQ, Limit of quantification. Accuracy and repeatability were calculated at 10 μg/L. LOD and LOQ were estimated by 3.3 (LOD) or 10 (LOQ) × standard deviation of the blank/slope of the calibration curve. ^a^ Coefficient of determination. ^b^ Relative standard deviation. ^c^ Relative error.

## Data Availability

Not applicable.

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
