# Peer review of "Examination of Microcystin Adsorption by the Type of Plastic Materials Used during the Procedure of Microcystin Analysis"

_toxins, 2022, doi:10.3390/toxins14090625_

Round 1

Reviewer 1 Report

The manuscript is a generally interesting work investigating the adsorption of 3 microcystin congeners in different plastic materials used in containers and filters during analytical processes. The term “microcystin” is incorrectly used throughout the manuscript to describe actually a group of numerous toxin congeners, so “microcystins” would be more suitable to use. One of the main problems identified in this research is the absence of control samples (that is blanks only with solvent) in each of the two sets of experiments, so the effect of the solvent used to dissolve the standard toxins (which is unknown to the reader, since standard preparation is not described; is it only water? or maybe methanol?) is not at all considered in the experimental design. Secondly, no data are provided on the quality control of the analysis method used to determine the 3 microcystin congeners, i.e. LOD, LOQ, linearity, recovery etc. in order to conclude whether the results provided are reliable. To my opinion the above limitations should be taken into consideration in the revised manuscript and: (1) include additional results of how blanks behave in each of the experiment sets (and in quantities resembling a routine MCs analysis in water samples); and (2) provide data on the quality control of the LC-MS/MS analysis method used for microcystins’ determination. The use of English is also quite poor, so the authors are advised to have their manuscript checked by a native English speaker or use a professional editing service before submitting a revised manuscript. Some additional points are also highlighted for revision, as follows:

General

- ‘Microcystis’ is a cyanobacterial genus name, it should be written in capital and italics (several instances in the text).

- The term ‘microcystin’ is incorrectly used since many microcystins exist. The authors should refer to either ‘microcystins’ in general, or mention the specific microcystin concerned in each case. For instance, in page 1, lines 27-29, the structure of ‘microcystin’ is described and 246 derivatives (which are either congeners or analogues, but not derivatives) are mentioned. It is better to use ‘microcystins’ to cover all cases. There are multiple instances in the text which should be revised accordingly.

Abstract

- Page 1, lines 6-7: revise the use of the terms ‘microcystis’ and ‘microcystin’ as indicated above.

- Page 1, lines 11-13: It should be clarified that this part refers to containers only - the term 'plastics' is used in both containers and membranes’ cases and it is confusing as written here.

1. Introduction

- Page 1, lines 26-27: revise the use of the terms ‘microcystis’ and ‘microcystin’ as indicated above.

- Page 1, lines 27-29: does this structure refer to all MCs? Please clarify.

- Page 1, lines 36-37: Which food ingredients are the authors referring to? The references provided do not refer to foods, only one of them refers to water.

- Page 1, lines 38-39: The WHO guidelines have been recently revised; the authors should be aware and refer to the newest standard of 2020 for microcystins’ contents in water (https://apps.who.int/iris/bitstream/handle/10665/338066/WHO-HEP-ECH-WSH-2020.6-eng.pdf), which differentiate between long-term (lifetime) and short-term exposure to MCs.

- Page 1, lines 43-45: “However, it has been reported that MC can be adsorbed on various types of plastics, such as poly-styrene (PS), polypropylene (PP) and polyvinyl chloride (PVC).”: Where has this been reported? At least one reference is needed here.

- Page 2, lines 50-51: “However, adding organic solvent during the sample preparation results in the sample dilution effects, which hinders the detection of MC in analytical samples.”: Please clarify what is meant by this? Does this work refer only to MCs analysis of water samples or other tissues as well? If so, it needs to be specified in the title that the work only refers to water samples’ analysis.

- Page 2, lines 52-54: Please rephrase, syntax is confusing.

- Page 2, lines 55-56: “…this study thoroughly examined the adsorption rate of MC by different plastic materials used for plastic containers and syringe filters.”: The authors should indicate only the MCs studied, the results cannot be generalized for all MCs.

2. Materials and Methods

- Page 2, line 59: Please provide details on the standards used (certified concentration if CRM or purity if not certified).

- Page 2, lines 74-75: place figure reference after "tubes", otherwise the reader expects to see the steel grinder in the figure and not the structure of the plastics.

- Page 2, line 76: use μm - not um, keep units consistent throughout the manuscript.

- Page 2, lines 78-81: Please clarify how the standard was prepared (in what solvent), since it was probably a mixture (or not?), and what was its initial concentration to achieve the final concentration indicated. Does this concentration refer to all three congeners? More detail is needed.

- Page 3, lines 91-92: Was the filter washed afterwards to obtain the remaining material? Also, how does this sample quantity of 500 μl simulate a normal analysis? Is this quantity equivalent to what is generally used in a routine microcystins' determination in water?

- Page 3, line 95: I assume the authors mean “Xevo” and not “Xeno”.

- Page 3, line 110: I assume the authors mean “Diego” and not “Dieago”.

3. Results

- Please describe how LOD and LOQ were determined and provide details on the method LOQ, LOD, recovery and linearity.

- Control samples should be included to provide solid results, please provide relevant data, the work is incomplete without examining the solvent effects.

- Page 4, Figure 2: It should be clear in the figure caption that this refers to containers – plastic materials are more or less the same in both containers and membrane filters.

4. Discussion

- Page 4, line 167: “hepatotoxic”: I assume the authors mean ‘hepatotoxicity’, but this is not the only type of toxicity caused by MCs, unless they refer to a specific one...

- Page 4, lines 168-169: if this was already tested in previous works, why was it repeated in this one? And to what MCs do these works refer to? E.g. [27] refers only to MC-LR and this should be clear in the discussion, to argue for the originality of the present work.

- Page 5, lines 181-182: Please rephrase, syntax is confusing.

- Page 5, line 183: “hydrophobic microcystin analogs”: which are these specifically?

- Page 5, lines 200-202: “… PP has a higher adsorption capacity than PET, and this result is explained by the rubbery state, crystallinity, and van der Waals interactions”: In which compounds has this been found? Is it MCs or is it other compounds?

- Page 5, lines 202-204: Please rephrase, syntax is confusing.

- Page 5, line 207: “it is judged”: most probably “it is assumed”.

- Page 5, lines 216-217: “In the case of PFA (Fig. 2), a relatively desirable recovery rate of ≥75.3% was confirmed in a short time (1 h)”: this is about containers, not membrane filters, please rephrase to demonstrate why it has a place here where syringe filtration is discussed.

- Page 6, lines 249-250: Ref [27] specifically refers to MC-LR not microcystins in general, please revise the text for clarity and indicate the exact material of the micropipette mentioned.

- Page 6, lines 254-256: Please rephrase, syntax is confusing.

5. Conclusions

- Page 6, lines 260-261: Are the membranes non-plastics? This should probably be revised to "the effect of plastics used in containers and membranes" or something equivalent… Please check the whole manuscript, this terminology issue appears throughout the text.

Reviewer 2 Report

This manuscript examined the recovery rate of three microcystin variant (RR, LR and YR) microcystins from six plastics for containers and eight plastics for membrane filters.

In general, this topic is important and interesting. However, there are still some issues need to be addressed. The detailed suggestions and comments are as follows:

1. L7

Change “microcystis” to “Microcystis”.

2. L29

“about 246 derivatives have been reported”

More than 279 derivatives have been reported. See and cite the following paper.

Challenges of using blooms of Microcystis spp. in animal feeds: A comprehensive review of nutritional, toxicological and microbial health evaluation. https://doi.org/10.1016/j.scitotenv.2020.142319

3. L32-34

References are missing.

4. L43-45

References are missing.

5. Results

Figure 2

Results are shown by effect of six plastic materials on recoveries of a given MC variant as a figure. Add results for recoveries of three MC variants by a given plastic material as a figure. Then, six figures, including PS, PP, LDPE, HDPE, PET and PFA, would be added.

6. Table 1. Recovery rate of microcystin for eight membrane filters.

Significant changes are shown by three MC variants or eight membrane filters?

7. Table 1

Add figures for effect of eight membrane filters on recoveries of a given MC variant as a figure, then six figures, including LR, RR, YR, would be added. Also, Add figures for recoveries of three MC variants by a given membrane filter as a figure. Then, eight figures, including RC, PVDF, PTFE, CA, MCE, PES, NY and PP, would be added.

8. Discussion

Discussion on adsorption of MC by plastic materials and membrane filters should be separated and put in independent paragraphs.

9. Reference list

The style of references is not based on requirements of the journal Toxins. First names of authors should be abbreviative.

Round 2

Reviewer 1 Report

The manuscript has generally improved with this first revision. However, the use of the terms “microcystin” (MC) or “microcystins” (MCs) still remains quite problematic. Few other issues still need some attention, particularly related to newly added details. The text still requires some “polishing” as regards the use of English Points for further revision follow:

General

- Whole manuscript: All instances of “microcystins” should be replaced with “MCs” and not “MC”. Otherwise, in case e.g. MC-LR is mentioned, we’re talking about microcystins-LR (which does not exist, of course). So, please revise the use of the terms Microcystins (MCs) and microcystin (MC, for instance in MC-LR) throughout the manuscript text, not only in the indicative lines mentioned in the previous report. There needs to be some consistency throughout the manuscript, so read through and correct accordingly (also syntactically, MCs cannot be treated as singular). For instance, in lines 121 and 145 it is correctly used.

1. Introduction

- Page 1, line 34: “derivatives”: just a few lines up “congeners” was used – revise for consistency.

- Page 2, lines 51-52: “However, adding organic solvent during the sample preparation results in the sample dilution effects, which hinders the detection of MC in analytical samples.”: This is still unclear as written. Does the present work examine the effect of methanolic or aqueous solution of MCs on plastics? Because both water and methanol are mentioned in the standards preparation. What is the ratio of these two in the test solutions?

2. Materials and Methods

- Page 2, lines 61 and 76: How is it possible that the initial standard has a concentration of 10 μg/ml in methanol and the stock solutions have a concentration of 100 μg/ml in methanol? Can the authors explain how this was achieved? Also, it is not explained if the three congeners were mixed in one standard (multitoxin) solution, or three different standard solutions (one for each congener) were used for the experiments.

- Page 3, lines 117-118: Please explain how statistical significance (figure 3) was evaluated (what approach was used), only means and SD are mentioned.

3. Results

- Page 3, lines 121-122: Please rephrase, syntax is confusing.

4. Discussion

- Page 4, lines 198-199: MCs may cause hepatotoxicity, but may also have other effects (e.g. cardiotoxicity, gastrointestinal symptoms etc.). Please revise the text for clarity, as it refers to all MCs not a particular one (e.g. MC-LR).

References

- Referencing style needs some revision to comply with the ‘Toxins’ requirements (e.g. journal abbreviations, etc.).

- Ref. [27] is incorrectly referenced. The W.H.O. stands for World Health Organization, it is not an author name to use initials as such. Access link needs to be added.

Reviewer 2 Report

This manuscript examined the recovery rate of three microcystin variants (RR, LR and YR) from six plastic particles for containers and eight plastics for membrane filters.

The quality of manuscript has improved a lot during revisions. However, there are still some issues need to be addressed. The detailed suggestions and comments are as follows:

1. Introduction

L 51-52

“However, adding organic solvent during the sample preparation results in the sample dilution effects, which hinders the detection of MC in analytical samples.”

What do you mean by sample dilution effects?

2. Materials and Methods

L 61-62

“Microcystin (RR, LR and YR, 95.0%, 10 μg/mL in methanol, analytical standard) and formic acid were purchased from Sigma-Aldrich (Louis, USA).”

L 76-77

“Standard stock solutions for MC-LR, -RR and -YR were prepared at 100 μg/mL in methanol.”

How to get 100 μg/mL of MCs from 10 μg/mL?

3. L 80-86

“2.3. Adsorption of microcystins on the different plastic materials of storage container

Six different types of plastic particles, including PS, PP, LDPE, HDPE, PET and PFA from plastic bottles and tubes, were prepared by grinding the bottles and tubes (Fig.1A) using a steel grinder. The ground plastic particle was filtered through 400 μm sieve to remove small particles. Subsequently, the residue was filtered using 800 μm sieves to prepare a plastic particle ranging from 400-800 μm. Briefly, 30 mg of the plastic particle was added to 1.5 mL glass vial.”

Plastic particles ranging from 400-800 μm are not the same as plastic containers. Please revise descriptions of results in the manuscript and discuss the differences between particles and containers. This is one limitation of this study. Add some discussion about it.

4. L 116-118

“2.6. Statistical analysis

The experiment was conducted in triplicate, and results were presented as mean±SD. Statistical analysis was performed using GraphPad Prism version 5.03 (San Diego, CA)”

Which analysis method did you use?

5. Figure 2 and Supplementary Figure 2

Results of statistical analysis were not shown.

6. Figure 2 and Supplementary Figure 2

Why recovery rates at 6 h of some curves were higher than those at 1 h?

7. Reference list

The style of references is not based on requirements of the journal Toxins. Names of journals should be abbreviative. Check the reference style for the journal.
